# Intrinsic Dimension Estimation for Robust Detection of AI-Generated Texts

**Eduard Tulchinskii**[1], **Kristian Kuznetsov**[1], **Laida Kushnareva**[2], **Daniil Cherniavskii**[3],
**Sergey Nikolenko**[5], **Evgeny Burnaev**[1,3], **Serguei Barannikov**[1,4], **Irina Piontkovskaya**[2]

[1]Skolkovo Institute of Science and Technology, Russia;
[2]AI Foundation and Algorithm Lab, Russia;
[3]Artificial Intelligence Research Institute (AIRI), Russia;[4]CNRS, Université Paris Cité, France;
[5]St. Petersburg Department of the Steklov Institute of Mathematics, Russia

## Abstract

Rapidly increasing quality of AI-generated content makes it difficult to distinguish between human and AI-generated texts, which may lead to undesirable consequences for society. Therefore, it becomes increasingly important to study the properties of human texts that are invariant over different text domains and varying proficiency of human writers, can be easily calculated for any language, and can robustly separate natural and AI-generated texts regardless of the generation model and sampling method. In this work, we propose such an invariant for human-written texts, namely the intrinsic dimensionality of the manifold underlying the set of embeddings for a given text sample. We show that the average intrinsic dimensionality of fluent texts in a natural language is hovering around the value 9 for several alphabet-based languages and around 7 for Chinese, while the average intrinsic dimensionality of AI-generated texts for each language is $\approx 1.5$ lower, with a clear statistical separation between human-generated and AI-generated distributions. This property allows us to build a score-based artificial text detector. The proposed detector's accuracy is stable over text domains, generator models, and human writer proficiency levels, outperforming SOTA detectors in model-agnostic and cross-domain scenarios by a significant margin. We release code and data[1]

## 1 Introduction

Modern large language models (LLMs) generate human-looking texts increasingly well, which may also lead to worrisome consequences [Fagni et al., 2021, Adelani et al., 2020, Stokel-Walker, 2022]. Hence, the ability to detect AI-generated texts (*artificial text detection*, ATD) becomes crucial for media, education, politics, creative industries and other spheres of human social activities. A straightforward idea would be to train a classifier to detect artificial text; many such classifiers exist [Zellers et al., 2019, Gehrmann et al., 2019, Solaiman et al., 2019], but most of them are designed to detect samples of individual generation models, either using the model itself [Mitchell et al., 2023] or training on a dataset of its generations. This leads to poor generalization to new models and unknown data domains. Another idea, known as *watermarking*, is to inject some detectable artifacts into model generations; for instance, Kirchenbauer et al. [2023] propose to intentionally inject a statistical skew that can be detected in a text sample. However, later works showed that watermark detectors can be broken by adversarial attacks, e.g., by text perturbations or paraphrasing [He et al., 2023]. Since text generation is constantly evolving, Sadasivan et al. [2023] claim that perfect artificial text detection is impossible; Krishna et al. [2023] address this statement and propose a retrieval-based detector that could be implemented by text generation service providers: they should store the hash

---

[1]`github.com/ArGintum/GPTID`

37th Conference on Neural Information Processing Systems (NeurIPS 2023).

value of every text generated by their model and retrieve it by request. This approach works even for a perfect text generator indistinguishable from human writing, but it does not apply to publicly available models, and plenty of them already exist.

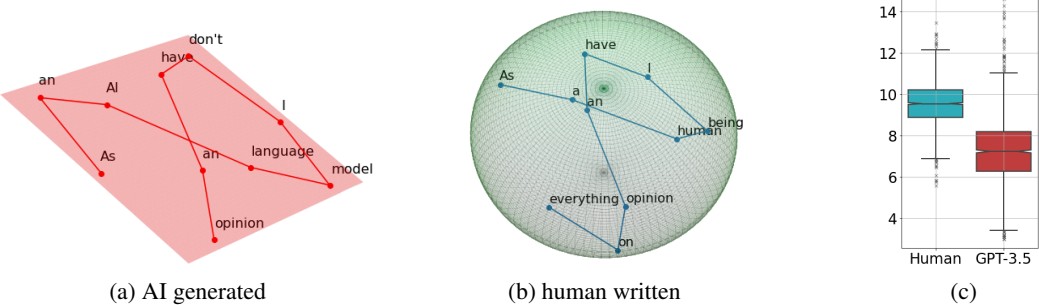

(a) AI generated        (b) human written        (c)

Figure 1: Real and artificial text have different intrinsic dimension: (a-b) idea; (c) actual results.

In this work, we show that the *intrinsic dimension* of text samples can serve as a helpful score function allowing to separate artificial and generated texts in a very general setting, without additional knowledge about the generator. The only assumption is that generation is good enough to create fluent grammatical samples of length $\approx 200$ words. We propose a method based on persistent homology dimension theory, which allows to estimate the dimension of text samples with high accuracy, and show that the proposed dimension-based classifier outperforms other artificial text detectors with a large margin in the general-purpose setup, for a very wide class of generators.

Many works have estimated the intrinsic dimension of data representations [Pope et al., 2021, Barannikov et al., 2021], neural network weights [Ansuini et al., 2019], or parameters needed to adapt to some downstream task [Aghajanyan et al., 2021], but these objects are very complex. Even if we are certain that a dataset fits into some surface in a high-dimensional feature space, it is not easy to estimate its dimension due to various kinds of noise (natural irregularities, measurement noise, numerical approximations) and the ambiguity of estimating a surface from a sparse set of points.

We estimate the geometry of every text sample as a separate object. Since texts generated by modern LLMs are fluent and usually do not contain grammatical, syntactical, or local semantic inconsistencies, we are interested in global sample geometry rather than properties that could be detected over short text spans. We show that the persistent dimension estimator provides an excellent way to deal with textual data: it turns out that real texts have a higher intrinsic dimension than artificial ones (Fig. 1). We propose an efficient method to implement the estimator and evaluate its classification ability in various settings, proving its robustness for artificial texts detection and showing that it works equally well across a number of different languages.

Our main contributions are thus as follows: (1) we propose to estimate the intrinsic dimensionality of natural language texts with the persistent homology dimension estimator and develop an efficient algorithm for computing it; (2) we show that the intrinsic dimension serves as a good score for artificial text detection for modern LLMs; in cross-domain and cross-model settings our method outperforms other general purpose classifiers by a large margin, is robust to adversarial attacks, and works for all considered languages; (3) we show that our text detector reduces the bias against non-native speakers in comparison to available ATD models; (4) we release a multilingual dataset of generations produced by GPT-3.5 and natural texts from the same domain in order to enable further ATD research. Below, Section 2 reviews related work, Section 3 introduces instrinsic dimension and its estimation with persistent homology, Section 4 applies it to artificial text detection, Section 5 presents our experimental evaluation, Section 6 discusses the limitations, and Section 7 concludes the paper.

## 2 Related work

Artificial text detection (ATD) becomes increasingly important with modern LLMs. GPT-2 [Radford et al., 2019] was accompanied by a work by Solaiman et al. [2019] on potential dangers and defences against them; the best ATD classifier there was based on supervised fine-tuning of RoBERTa [Liu et al.,

2019]. Supervised approaches can work well for other generative models and data domains [Krishna et al., 2023, Guo et al., 2023, He et al., 2023] but they do not generalize to other text domains, generation models, and even sampling strategies [Bakhtin et al., 2019, Solaiman et al., 2019]. In the zero-shot setting, Solaiman et al. [2019] threshold the average log-probability score of a sample calculated by some pretrained language model (LM). DetectGPT [Mitchell et al., 2023] treats log-probability calculation as a function, estimates its curvature in a small neighbourhood of the text, and shows that this curvature score is smaller for artificial texts (there are "flat" regions around them); however, DetectGPT needs the likelihood to come from the same LM as the sample.

We focus on developing an ATD model that could generalized to unseen text generation models and domains. Zellers et al. [2019] detect generated texts perfectly with a discriminator model built on top of the generator, but the quality drops significantly when the generator changes, even with supervised adaptation; a similar "model detects itself" setup was adopted by Mitchell et al. [2023]. Solaiman et al. [2019] show that a simple score-based approach by the likelihood score works well in the "model detects itself" setting but does not generalize to different generator and discriminator; as for transferability, they show that a supervised classifier generalizes well when it is trained on the output of a more complex model and transferred to a less complex one but not in the reverse direction. Bakhtin et al. [2019] consider different types of generalization: in-domain (train and test generators are the same), cross-corpus (train and text generators fine-tuned on different corpora), and cross-architecture (train and test generators have different architectures but the same training corpora); their model shows good in-domain generalization ability, handles relatively well cross-architecture generalization, but loses quality in cross-corpus generalization. Mitchell et al. [2023] demonstrate the stability of their method over text domains compared to supervised models, which are better on in-domain data but lose efficiency in a cross-domain setting. Finally, Krishna et al. [2023] show all methods failing dramatically against the DIPPER paraphrase attack (except for a lower-performing approach developed for text quality ranking [Krishna et al., 2022]). We also note a work by Liang et al. [2023] who show the bias of artificial text detectors against non-native speakers and show that all existing detectors can be broken by generating texts with controllable complexity.

Geometrical and topological methods have shown their usefulness for analysing the intrinsic dimensionality of data representations. Some works focus on data manifolds [Pope et al., 2021, Barannikov et al., 2021] and others consider hidden representations and other parts of neural networks and investigate through the lens of intrinsic dimensionality. Li et al. [2018] define the intrinsic dimension of objective landscape by tracking the subspace dimension and performance of a neural network, while Zhu et al. [2018] use manifold dimension directly to regularize the model. Ansuini et al. [2019] apply TwoNN to internal representations in CNNs and establish a connection to the model's generalization ability. Birdal et al. [2021] show that the generalization error of these models can be bounded via persistent homology dimension. Vision transformers were also investigated by Xue et al. [2022] and Magai and Ayzenberg [2022]. Moreover, intrinsic dimensionality has been connected to the generalization of Transformer-based LLMs [Aghajanyan et al., 2021]. Valeriani et al. [2023] analyze the intrinsic dimensionality of large Transformer-based models. Topological properties of the inner representations of Transformer-based models [Vaswani et al., 2017], including BERT [Devlin et al., 2019] and HuBERT [Hsu et al., 2021], have been successfully applied for solving a wide variety of tasks, from artificial text detection [Kushnareva et al., 2021] and acceptability judgement [Cherniavskii et al., 2022] to speech processing [Tulchinskii et al., 2023].

## 3 Intrinsic dimension and persistent homology dimension

Informally speaking, the intrinsic dimension of some subset $S \subset \mathbb{R}^n$ is the number of degrees of freedom that a point moving inside $S$ has. This means that in a small neighbourhood of every point, the set $S$ can be described as a function of $d$ parameters, $d \leq n$, and this number cannot be reduced. This idea is formalized in the notion of a $d$-dimensional *manifold* in $\mathbb{R}^n$: it is a subset $M \subset \mathbb{R}^n$ such that for every point $x \in M$ there exists an open neighborhood which is equivalent to an open ball in $\mathbb{R}^d$ for some value $d$. Importantly, if $M$ is a connected set then $d$ should be the same for all its points, so we can talk about the dimension of the entire manifold.

Data representations often use excessive numbers of features, some of which are highly correlated. This overparametrization has been noticed many times [Hein and Audibert, 2005, Kuleshov et al., 2017, Pope et al., 2021], and the idea that real data lies (approximately) on some low-dimensional manifold in the feature space is known as the *manifold hypothesis* [Goodfellow et al., 2016]. However,

there are obstacles to estimating the intrinsic dimension of a dataset. First, a real dataset can be a combination of sets of different dimensions. Second, data can be noisy and may contain outliers. Moreover, real data can have a complicated hierarchical structure, so different approximation methods lead to different intrinsic dimension values. For an analogy, consider the observations of a single spiral galaxy that consists of separate points (stars, planets etc.) but forms a compact 3-dimensional manifold. At some level of approximation the galaxy looks like a disk, which is 2-dimensional, but if we take a closer look we discover the structure of a 3-dimensional core and basically 1-dimensional arms. Moreover, if we add observations over time, the dataset will consist of 1-dimensional trajectories of individual points that exactly correspond to well-defined mathematical trajectories (the noise here comes only from measurement errors); these trajectories form an approximate 3-dimensional cylinder in 4-dimensional space with a much higher level of noise around its borders. As a result, the dimension of the entire object can be estimated by any number from 1 to 4 depending on the detector's sensitivity to noise and outliers, preference for global or local features, and the way to average the values of the non-uniform distribution of the points.

Thus, it is natural that there exist several different methods for intrinsic dimension (ID) estimation, and we have to choose the one most suitable for the task at hand. For example, many ID estimators are based on constructing a global mapping of the data into a lower-dimensional linear subspace, with either linear projection (e.g., PCA), kernel-based methods, or distance-preserving nonlinear transformations. However, in our preliminary experiments these types of dimension estimation seemed to be losing information that was key for artificial text detection.

We focus on the *persistent homology dimension* estimator (PHD) [Schweinhart, 2021], which belongs to the class of *fractal dimension* approaches. Consider a ball of radius $r$ inside a $d$-dimensional manifold $M$. As $r$ grows, the volume of the ball increases proportionally to $r^d$. Let $x_1, ..., x_N$ be points uniformly sampled from $M$. Then the expected number of points in a ball of radius $r$ also changes as $r^d$ with $r$. Naturally, real datasets usually do not correspond to a uniform distribution of points, but this issue can be overcome by considering the asymptotic behaviour of the number of points in an $r$-ball as $r \to 0$. In this case, it suffices for the data distribution to be smooth and therefore close to uniform in the neighbourhood of every point. Accurate straightforward estimation of $d$ based on the above observation is not sample-efficient but there exist several approximate approaches, including the MLE dimension that evaluates the data likelihood [Levina and Bickel, 2004], the TwoNN dimension that uses the expected ratio of distances from a given point to its two nearest neighbours [Facco et al., 2017], and MADA [Farahmand et al., 2007] that uses the first order expansion of the probability mass function. We also report MLE-based results as its performance is comparable to PHD in some tasks.

We propose to use *persistence homology dimension* (PHD) that has several appealing properties compared to other fractal intrinsic dimension estimators. First, the above methods operate locally while PHD combines local and global properties of the dataset. Second, according to our experiments, this method is sample-efficient and redundant to noise (see below). Third, it has a solid theoretical background that connects topological data analysis, combinatorics, and fractal geometry [Adams et al., 2020, Birdal et al., 2021, Jaquette and Schweinhart, 2020, Schweinhart, 2021].

The formal definition of PHD is based on the concept of *persistent homology* for a set of points in a metric space, which is a basic notion of *topological data analysis* (TDA) [Chazal and Michel, 2017, Barannikov, 1994, 2021]. TDA aims to recover the underlying continuous shape for a set of points by filling in the gaps between them that are smaller than some threshold $t$ and studying the topological features of the resulting object as $t$ increases. Each persistent homology $\mathrm{PH}_i$ in a sequence $\mathrm{PH}_0, \mathrm{PH}_1, \ldots$ is defined by the set of *topological features* of dimension $i$: 0-dimensional features are connected components, 1-dimensional features are non-trivial cycles, 2-dimension features are tunnels, etc. For each feature we calculate its "lifespan", a pair $(t_{\mathrm{birth}}, t_{\mathrm{death}})$, where $t_{\mathrm{birth}}$ is the minimal threshold where the feature arises, and $t_{\mathrm{death}}$ is the threshold where it is destroyed.

Following Adams et al. [2020], we introduce the persistent homology dimension as follows. Consider a set of points $X = \{x_1, \ldots, x_N\} \subset \mathbb{R}^n$. We define the $\alpha$-*weighted sum* as $E_\alpha^i(X) = \sum_{\gamma \in \mathrm{PH}_i(X)} |I(\gamma)|^\alpha$, where $I(\gamma) = t_{\mathrm{death}}(\gamma) - t_{\mathrm{birth}}(\gamma)$ is the lifespan of feature $\gamma$. For $i = 0$, $E_\alpha^i$ can be expressed in terms of the minimal spanning tree (MST) of $X$: its edges map to lifespans of 0-dimensional features $\gamma \in \mathrm{PH}_0(X)$ [Bauer, 2021, Birdal et al., 2021]. Thus, the definition of $E_\alpha^0(X)$ is equivalent to $E_\alpha^0(X) = \sum_{e \in \mathrm{MST}(X)} |e|^\alpha$, where $|e|$ is the length of edge $e$.

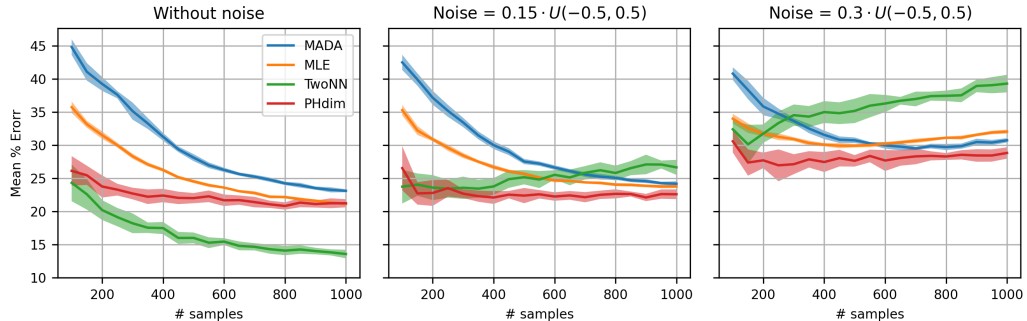

Figure 2: A comparison of ID estimators with noise on artificial datasets; lower is better.

There is a classical result on the growth rate of $E_\alpha^0(X)$ [Steele, 1988]: if $x_i$, $0 < i < \infty$ are independent random variables with a distribution having compact support in $\mathbb{R}^d$ then with probability one $E_\alpha^0(X) \sim Cn^{\frac{d-\alpha}{d}}$ as $n \to \infty$, where equivalence means that the ratio of the terms tends to one. It shows that $E_\alpha^0$ tends to infinity with $N$ if and only if $\alpha < d$. Now one can define the intrinsic dimension based on MST as the minimal value of $\alpha$ for which the score is bounded for finite samples of points from $M$ [Schweinhart, 2021]:

$$\dim_{\mathrm{MST}}(M) = \inf\{d \mid \exists C \text{ such that } E_d^0(X) \leq C \text{ for every finite } X \subset M\},$$

and a PH dimension as

$$\dim_{\mathrm{PH}}(M) = \inf\{d \mid \exists C \text{ such that } E_d^0(X) \leq C \text{ for every finite } X \subset M\}.$$

We now see that $\dim_{\mathrm{MST}}(M) = \dim_{\mathrm{PH}}(M)$ for any manifold $M$. This fact, together with the growth rate result above, provides a sample-efficient way to estimate $\dim_{PH}(M)$ [Birdal et al., 2021]: sample subsets $X_{n_i} = \{x_1, \dots, x_{n_i}\} \subset M$ of $n_i$ elements for a growing sequence of $n_i$, for every subset find its MST and calculate $E_\alpha^0(X_{n_i})$, and then estimate the exponent of the growth rate of the resulting sequence by linear regression between $\log E_\alpha^0(X_{n_i})$ and $\log n$, since we know that $\log E_\alpha^0(X_{n_i}) \sim (1 - \frac{\alpha}{d}) \log n_i + \tilde{C}$ as $n_i \to \infty$.

Next, we show empirically that our method of ID estimation via PHD approximates the real dimension of a manifold well and is well suited for the conditions mentioned earlier: presence of noise and small number of samples. To compare with other ID estimators, we utilize a benchmark by Campadelli et al. [2015] designed specifically for the evaluation of ID estimation methods and used the *scikit-dimensions* library [Bac et al., 2021] with efficient implementations of 12 different approaches to ID estimation, popular for different tasks. We evaluated many of these approaches on artificial datasets from Bac et al. [2021], 1000 samples each, without noise. Choosing three "winners"—MLE, TwoNN, and MADA,—we have evaluated their sample efficiency and noise tolerance in comparison with our implementation of the PHD estimator. Fig. 2 shows the results: PHD is the only method tolerant to noise, and it does not degrade when data is scarce. It outperforms all other methods in the noisy setup for any sample size. The second-best method is MLE, which performs relatively well on small samples (200–500) in noisy settings and has a small variance. Below we will show that as a result, MLE is also applicable to artificial text detection, but it lags a little behind PHD on average.

## 4 Methodology

We consider consistent text samples of medium size, with length $\approx 300$ tokens; we assume that each text contains a complete thought or is devoted to a single topic. We estimate the dimension of each text sample, considering it as a separate manifold. To do this, we obtain contextualized embeddings for every token in the text by a pretrained Transfromer encoder. In our experiments, we use RoBERTa-base [Liu et al., 2019] for English and XLM-R [Goyal et al., 2021] for other languages. Each embedding is a numerical vector of a fixed length, so we view it as a point in the Euclidean space. We drop the artificial first and last tokens (<CLS> and <SEP>) and evaluate the persistent homology dimension of the resulting point cloud using the growth rate theorem (see Section 3).

Table 1: Intrinsic dimensions of English texts of different genres.

| | Wikipedia articles | Fiction stories (Reddit) | Question answering (Stack Exchange) |
|---|---|---|---|
| PHD | $9.491 \pm 1.010$ | $9.212 \pm 1.288$ | $9.594 \pm 1.29$ |
| MLE | $11.827 \pm 0.768$ | $11.553 \pm 1.197$ | $12.131 \pm 1.004$ |

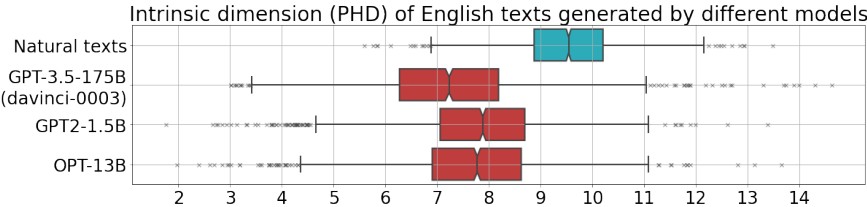

Figure 3: Boxplots of PHD distributions for different generative models in comparison to human-written text on Wikipedia data. Embeddings are obtained from RoBERTa-base.

Given a set of points $S$, $|S| = n$, we first sample subsets $S_i \subset S$, $i = 1, \ldots, k$ whose sizes $n_1, \ldots, n_k$ are uniformly distributed in $[1, n]$. For each $S_i$ we calculate its persistent score $E_0^1(S_i)$ (just $E(S_i)$ below); this can be done with a classical MST algorithm in linear time. Then we prepare a dataset consisting of $k$ pairs $D = \{(\log n_i, \log E(S_i))\}$ and apply linear regression to approximate this set by a line. Now the dimension $d$ can be estimated as $\frac{1}{1-\kappa}$, where $\kappa$ is the slope of the fitted line.

In general, our method for PHD calculation is similar to the the computational scheme proposed by Birdal et al. [2021]. But since we are dealing with sets that are much smaller and less uniformly distributed, their algorithm becomes unstable, with variance up to 35% of the value from different random seeds; moreover, if one of the subsets $S_i$ slips into a local density peak and has an unusually low persistence score, the algorithm may even produce a meaningless answer (e.g., negative $d$).

To overcome this issue, we add several rounds of sampling and averaging to improve the stability of calculation. We estimate the expectation $\mathbb{E}_{s \subset S, |s|=n_i}[E(s)]$ for a given $n_i$ instead of direct calculation of $E(S_i)$ for a single sample. For that, we perform the whole process of computing $d$ several times, averaging the results. Details of our sampling schema can be found in the Appendix.

Finally, we construct a simple single-feature classifier for artificial text detection with PHD as the feature, training a logistic regression on some dataset of real and generated texts.

## 5   Experiments

**Datasets**. Our main dataset of human texts is Wiki40b [Guo et al., 2020]. We measured intrinsic dimension of fiction stories on the target split of the WritingPrompts dataset [Fan et al., 2018], a collection of short texts written by Reddit users. For multilingual text detection experiments, we generated a new WikiM dataset for 10 languages by GPT3.5-turbo (ChatGPT). We use the header and first sentence from a Wikipedia page as the prompt and ask the model to continue. In cross-domain and paraphrase robustness experiments, we use Wiki and Reddit datasets (3k samples each) [Krishna et al., 2023] that use two consecutive sentences (Wiki) or the question (Reddit) as a prompt and generate texts by GPT2-XL, OPT13b, and GPT3.5 (text-davinci-003). Following their pipeline for Reddit, we have also generated a StackExchange dataset by GPT3.5 (text-davinci-003) as the third domain. We select questions posted after 2019-08-01 from non-technical categories, where both question and answer have rating more then 10, and clean them removing HTML artifacts. In order to assess the bias in our estimator, we use the data provided by Liang et al. [2023].

**Intrinsic dimensionality of real and generated texts**. First, we observe an intriguing fact: the intrinsic dimension of natural texts is mostly concentrated between values **9** and **10**, while the dimension of generated texts is lower and is approximately equal to **8**, regardless of the generator. This is illustrated in Figure 3. Table 1 shows that this value is stable across different text genres but slightly varies for different languages: it is approximately equal to $\mathbf{9 \pm 1}$ for most European languages,

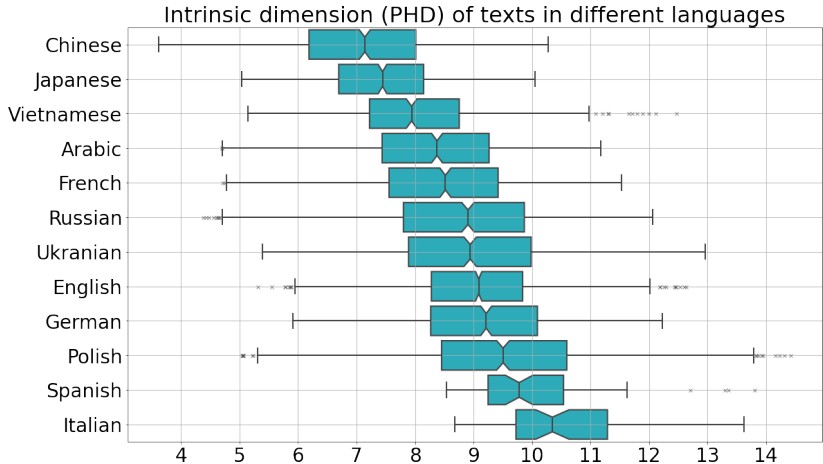

Figure 4: Boxplots of PHD distributions in different languages on Wikipedia data. Embeddings are obtained from XLM-RoBERTa-base (multilingual).

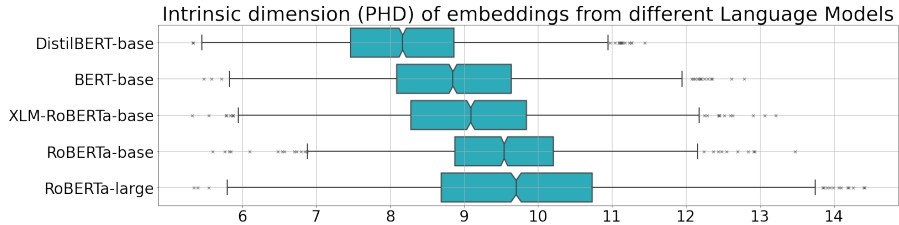

Figure 5: Boxplots of PHD distributions obtained by different LMs on English Wikipedia data.

slightly larger for Italian and Spanish ($\approx 10 \pm 1$), and lower for Chinese and Japanese ($\approx 7 \pm 1$); details are shown in Fig. 4. But we always observe a clear difference between this distribution and generated texts on the same language (see Appendix for more experiments).

Next, we check how the PHD estimation depends on the base model that we use for text embedding calculation. Fig. 5 demonstrates that PHD changes slightly with the change of the base LM, decreasing for models with fewer parameters. RoBERTa-base embeddings provide the best variance for PHD estimation, so we use this model for all further experiments in English, and XLM-R of the same size for multilingual experiments.

**Artificial text detection**. We show that intrinsic dimension can lead to a robust method of artificial text detection. In all experiments below, we use the one-feature thresholding classifier (see Section 4).

**Comparison with universal detectors**. First, we show that our detector is the best among general-purpose methods designed to detect texts of any domain, generated by any AI model, without access to the generator itself. Such methods are needed, e.g., for plagiarism detection. To be applicable in real life, the algorithm should provide high artificial text detection rate while avoiding false accusations of real authors. Besides, it should be resistant to adversaries who transform the content generated by popular AI models to reduce the chance to be caught.

Here we adopt the experimental settings by Krishna et al. [2023] and use the baseline results presented there. We compare PHD and MLE with two general-purpose detectors: GPTZero [Tian, 2023], targeted to detect the texts generated by contemporary LLMs (GPT-3, GPT-4, ChatGPT, BARD), and OpenAI detector [OpenAI, 2023] announced together with the ChatGPT model in order to reduce its expected social harm. Our third baseline is DetectGPT [Mitchell et al., 2023], which is a state of the art thresholding classifier that evaluates text samples by the probability curvature obtained via the generator model. It works best when the base model coincides with the generator model ("model detects itself") but the authors claim that it can generalize to cross-model setup with reasonable quality. RankGen [Krishna et al., 2022] is a method originally developed for ranking hypotheses during text generation; it demonstrates a surprising ability to handle adversarial attacks.

Table 2: Artificial text detection (accuracy at 1% FPR) for open-ended generation using Wikipedia prompts. DIPPER was run with Lex=60, Order=60.

| Generator | Existing Solutions | | | | Our methods | |
| | DetectGPT | OpenAI | GPTZero | RankGen | PHD | MLE |
|---|---|---|---|---|---|---|
| GPT-2 | 70.3* | 21.6 | 13.9 | 13.5 | **25.2** | 23.8 |
| + DIPPER | 4.6 | 14.8 | 1.2 | **28.5** | 27.6 | 19.7 |
| OPT | 14.3 | 11.3 | 8.7 | 3.2 | **28.0** | 26.7 |
| + DIPPER | 0.3 | 10.0 | 1.0 | 13.5 | **30.2** | 22.1 |
| GPT-3.5 | 0.0 | 30.0 | 7.1 | 1.2 | 40.0 | **46.7** |
| + DIPPER | 0.0 | 15.6 | 1.8 | 7.3 | **41.2** | 33.3 |

Following Krishna et al. [2023], we report the detection accuracy with false positive rate (FPR) fixed at 1%. Table 2 shows that our PHD-based classifier outperforms all baselines with a large margin: +10% for GPT-3.5, +14% for OPT. Note that DetectGPT uses GPT-2 as the base model, which explains its results for GPT-2. PHD is also invulnerable to the DIPPER paraphrasing attack [Krishna et al., 2023]. When generated texts are transformed by DIPPER, they lose some characteristic features of the generator, which causes a dramatic drop in quality for most detectors; but for the PHD classifier the accuracy of artificial text detection even increases slightly after this perturbation. Interestingly, the MLE dimension estimator also works quite well for this task, and even achieves 6% better detection for GPT-3.5 generations; but its adversarial robustness is significantly worse.

**Cross-domain and cross-model performance**. Table 3 shows that our ID estimation is stable across text domains; consequently, our proposed PHD text detector is robust to domain transfer. We compare the cross-domain ability of PHD with a supervised classifier obtained by fine-tuning RoBERTa-base with a linear classification head on its $CLS$ token, a supervised classification approach used previously for artificial texts detection with very high in-domain accuracy [Solaiman et al., 2019, Guo et al., 2023, He et al., 2023]. We split data into train / validation / test sets in proportion 80%/10%/10%. Table 3 reports the results of the classifier's transfer between three datasets of different text styles—long-form answers collected from Reddit, Wikipedia-style texts, and answers from StackExchange—using data generated by GPT-3.5 (text-davinci-003). Although supervised classification is virtually perfect on in-domain data, it fails in cross-domain evaluation, while the PHD classifier is not influenced by domain transfer. On average, the PHD classifier slightly outperforms the supervised baseline, while being much more stable. Table 3 also reports cross-model transfer ability, where the classifier is trained on the output of one generation model and tested on another. We consider generations of GPT-2, OPT, and GPT-3.5 (text-davinci-003) in the *Wikipedia* domain and observe that the PHD classifier, again, is perfectly stable. This time, RoBERTa-base supervised classifier handles the domain shift much better and outperforms PHD on average, but it has a higher cross-domain generalization gap. This means that we can expect the PHD classifier to be more robust to entirely new AI models.

**PHD-based classification for other languages**. Table 4 presents the results of PHD-based artificial text detection for Wikipedia-style texts generated by ChatGPT in 10 languages. Text embeddings were obtained with XLM-RoBERTa-base, the multilingual version of RoBERTa. As quality metric we report the area under ROC-curve (ROC-AUC). We see that both ID classifiers provide solid results for all considered languages, with the average quality of 0.78 for PHD and 0.8 for MLE; MLE performs better for almost all languages. The worst quality is on Chinese and Japanese (PHD 0.71 and 0.74, MLE 0.65 and 0.75 respectively), the best is for Spanish and Italian (PHD 0.83, MLE 0.85 for both). Note that the best and worst classified languages are those with the largest and smallest ID values in Fig. 4; we leave the investigation of this phenomenon for further research.

**Non-native speaker bias**. Finally, we show how our model helps to mitigate the bias present in ML-based artificial text detectors. We follow Liang et al. [2023] who demonstrate that current artificial text detectors are often too hard on texts written by non-native speakers. We use OpenAI and GPTZero as the baselines (see Appendix for more results) and PHD and MLE classifiers, choosing the thresholds was chosen on data unrelated to this task, as the equal error classifier on introductions of Wikipedia articles (real vs GPT-3.5-turbo) where it achieved EER of 26.8% for PHD and 22.5% for MLE. On the left, Fig. 6 shows false positive rates (FPR) for three sets of student essays: TOEFL essays by non-native speakers (red), same texts processed by GPT-4 asked to improve the text (grey),

Table 3: Cross-domain and cross-model accuracy of PHD and RoBERTa-based classifiers on data from three different domains and three different models; classes are balanced in training and evaluation.

| | **RoBERTa-cls** | | | **Intrinsic Dimension (PHD)** | | |
|---|---|---|---|---|---|---|
| Train \ Eval | Wikipedia | Reddit | StackExchange | Wikipedia | Reddit | StackExchange |
| Wikipedia | 0.990 | 0.535 | 0.690 | 0.843 | 0.781 | 0.795 |
| Reddit | 0.388 | 0.997 | 0.457 | 0.855 | 0.776 | 0.773 |
| StackExchange | 0.525 | 0.473 | 0.999 | 0.834 | 0.778 | 0.800 |
| Train \ Eval | GPT2 | OPT | GPT3.5 | GPT2 | OPT | GPT3.5 |
| GPT2 | 0.992 | 0.993 | 0.933 | 0.769 | 0.759 | 0.832 |
| OPT | 0.988 | 0.997 | 0.967 | 0.769 | 0.763 | 0.837 |
| GPT3.5 | 0.937 | 0.982 | 0.990 | 0.759 | 0.757 | 0.843 |

Table 4: Quality of artificial text detection in different languages (ROC-AUC) for ChatGPT text.

| Language: | cn-zh | en | fr | de | it | jp | pl | ru | es | uk |
|---|---|---|---|---|---|---|---|---|---|---|
| PHD | 0.709 | 0.781 | 0.790 | 0.767 | 0.831 | 0.737 | 0.794 | 0.777 | 0.833 | 0.768 |
| MLE | 0.650 | 0.770 | 0.804 | 0.788 | 0.852 | 0.753 | 0.850 | 0.816 | 0.853 | 0.821 |

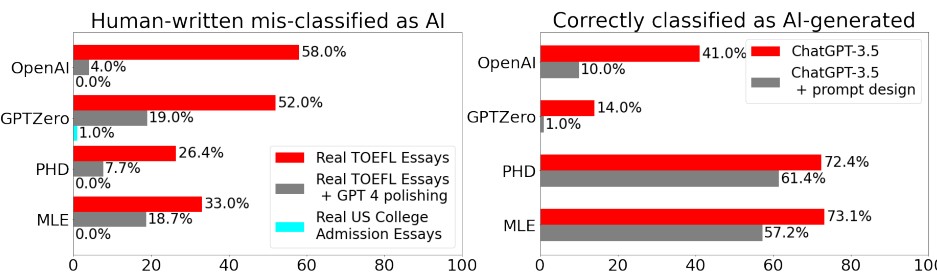

Figure 6: Comparison of GPT detectors in non-standard environment. Left: bias against non-native English writing samples (the lower is better). Right: effect of the prompt design on performance (the higher is better).

and native speakers (blue). First, blue bars are almost invisible for all detectors because the FPR for a native speaker is very small ($< 1\%$) while non-native speakers can be wrongly accused by OpenAI and GPTZero in $58\%$ and $52\%$ of the cases respectively. The PHD classifier reduces this discriminating rate by 2x, showing FPR $26\%$ for non-native speakers. After GPT-4 polishing, this rate further decreases to $7.7\%$ compared to $19\%$ for GPTZero. Interestingly, OpenAI also deals with GPT-4 polished texts suprisingly well, its FPR drops by 15x. The MLE detector also demonstrates less biased behaviour compared to baselines, but worse than PHD.

On the right, Fig. 6 shows the true positive rates (TPR) of these methods on essays generated by ChatGPT. Red bars show that our classifiers greatly outperform baselines. Grey bars demonstrate the robustness of ID detectors to changes in generation style via prompt design. If an adversary asks ChatGPT to generate a text with some predefined level of complexity ("use simple words", or "more complex words"), baseline systems fail to correctly recognize such texts while both ID classifiers still yield high detection rates.

**Analysis of edge cases**. We noticed an interesting tendency among the human-written passages with the lowest ID (misclassified as AI-generated). It seems that most of these examples contain a lot of addresses, geographical names, or proper nouns; some texts, however, are typical but very short. We hypothesize that it can be related to an unusually high number of rare tokens or numbers in the text and that PH dimension estimation is overall less correct on short texts. Besides, Davinci-generated texts with the highest ID (misclassified as human-written with the most certainty) are also often quite short. We leave a more comprehensive analysis of such failure cases for future work.

We provide examples of both types of misclassified texts in Appendix E. Moreover, we show bar plots of the PHD of artificially created texts made from random tokens and texts composed by repeating the

same token in Figure 11. One can see that the simplest samples have the lowest ID, and the highest values of ID correspond to completely random texts. This supports the general understanding of ID as the number of degrees of freedom in the data.

# 6 Limitations and broader impact

We see three main limitations of our method. First, it is stochastic in nature. PH dimensions of texts from the same generator vary widely, and the estimation algorithm is stochastic as well, which adds noise, while rerolling the estimation several times would slow down the method. Second, "out of the box" our method can detect only "good" (fluent) generators with a relatively small temperature of generation. The PH dimension of "bad" or high-temperature generators is actually higher on average than for real texts, so the detector will need to be recalibrated. Third, we have only evaluated our approach on several relatively high-resource languages and we do not know how the method transfers to low-resource languages; this is a direction for future work. Nevertheless, our method provides a new tool for recognizing fake content without discriminating non-native speakers, which is also much more robust to model change and domain change than known tools.

# 7 Conclusion

In this work, we have introduced a novel approach to estimating the intrinsic dimension of a text sample. We find that this dimension is approximately the same for all human-written samples in a given language, while texts produced by modern LLMs have lower dimension on average, which allows us to construct an artificial text detector. Our comprehensive experimental study proves the robustness of this classifier to domain shift, model shift, and adversarial attacks. We believe that we have discovered a new interesting feature of neural text representations that warrants further study.

## Acknowledgments and Disclosure of Funding

The work of Evgeny Burnaev was supported by the Russian Foundation for Basic Research grant 21-51-12005 NNIO_a.

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

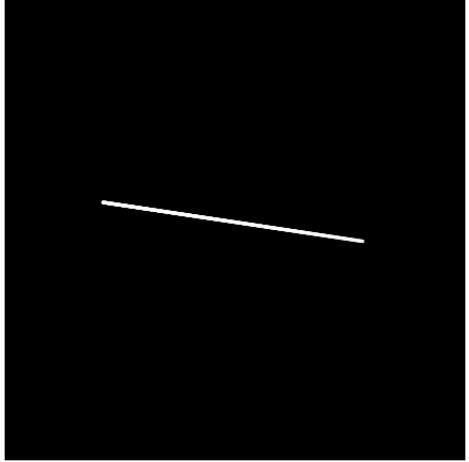

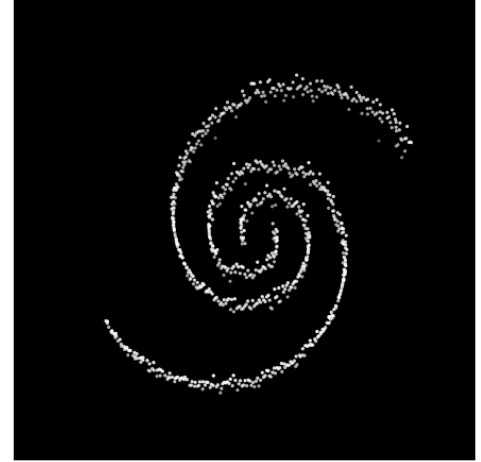

(a) Point cloud concentrated around a straight line; $PHD \approx 1.0$

(b) Point cloud concentrated around two curved intertwined lined; $PHD \approx 1.5$

Figure 7: Example of the difference between intrinsic dimensionality (PHD) of point clouds for different geometric shapes. The point cloud concentrated around a more complex structure (7b) has larger PHD than a point cloud concentrated around a simpler structure (7a).

# A    Theory

## A.1    Formal definitions of persistent homologies

A persistent homology is a sequence of homology groups and linear maps parameterized by a filtration value. Formally speaking, given a filtered chain complex $(K, \partial)$ with filtration values $\lambda_1 < \lambda_2 < \ldots < \lambda_n$, we have a sequence of chain complexes: $K_{\lambda_0} \subseteq K_{\lambda_1} \subseteq K_{\lambda_2} \subseteq \ldots \subseteq K_{\lambda_n} = K$. For each $\lambda_j$, the $i$-th homology group $H_i(K_{\lambda_j})$ denotes the factor vector space $H_i(K_{\lambda_j}) = \ker \partial|_{K_{\lambda_j}^{(i)}} / \operatorname{im} \partial|_{K_{\lambda_j}^{(i)}}$. The inclusion $K_{\lambda_j} \subseteq K_{\lambda_{j+r}}$ induces a linear map $f_{j,j+r} : H_i(K_{\lambda_j}) \to H_i(K_{\lambda_{j+r}})$. By definition, the persistent homology $\mathrm{PH}_i$ of the filtered chain complex is the collection of homology groups $H_i(K_{\lambda_j})$ and linear maps between them: $\mathrm{PH}_i = \{H_i(K_{\lambda_j}), f_{j,j+r}\}_{j,r}$.

By the structure theorem of persistent homology, a filtered chain complex $(K, \partial)$ is decomposed into the unique direct sum of standard filtered chain complexes of types $I(b_p, d_p)$ and $I(h_p)$, where $I(b, d)$ is the filtered complex spanned linearly by two elements $e_b, e_d$, $\partial e_d = e_b$ with filtrations $e_b \in I_b^{(i)}$, $e_d \in I_d^{(i+1)}$, $b \leq d$ and $I(h)$ is the filtered complex spanned by a single element $\partial e_h = 0$ with filtration $e_h \in I_h^{(i)}$ [Barannikov, 1994]. This collection of filtered complexes $I(b_p, d_p)$ and $I(h_p)$ from the decomposition of $K$ is called the $i$th *Persistence Barcode* of the filtered complex $K$. It is represented as the multiset of the intervals $[b_p, d_p]$ and $[h_p, +\infty)$. In Section 3, when we speak loosely about the persistent homology $\mathrm{PH}_i$, we actually mean the $i$th persistence barcode. In particular, the summation $\sum_{\gamma \in \mathrm{PH}_i}$ is the summation over the multiset of intervals constituting the $i$th persistence barcode.

## A.2    Equivalence between PH$_0$(S) and MST(S)

For the reader's convenience, we provide a sketch of the equivalence between the $0$th persistence barcodes and the set of edges in the minimal spanning tree (MST).

First, recall the process of constructing the $0$-dimensional persistence barcode [Adams et al., 2020]. Given a set of points $S$, we consider a simplicial complex $K$ consisting of points and all edges between them (we do not need to consider faces of higher order to compute $H_0$): $K = \{S\} \cup \{(s_i, s_j)|s_i, s_j \in S\}$. Each element in the filtration $K_{\lambda_0} \subseteq K_{\lambda_1} \subseteq K_{\lambda_2} \subseteq \ldots \subseteq K_{\lambda_n} = K$ contains edges shorter than the threshold $\lambda_k$: $K_{\lambda_k} = \{S\} \cup \{(s_i, s_j)|s_i, s_j \in S, ||s_i - s_j|| < \lambda_k\}$. The $0$-th persistence barcode is the collection of lifespans of $0$-dimensional features, which correspond to connected

components, evaluated with growths of the threshold $\lambda$. Let us define a step-by-step algorithm for evaluating the features' lifespans. We start from $\lambda = 0$, when each point is a connected component, so all the features are born. We add edges to the complex in increasing order. On each step, we are given a set of connected components and a queue of the remaining edges ordered by length. If the next edge of length $\lambda$ connects two connected components to each other, we claim the *death* of the first component and add a new lifespan $(0, \lambda)$ to the persistence barcode; otherwise, we just remove the edge from the queue since its addition to the complex does not influence the 0th barcode.

Now we can notice that this algorithm corresponds exactly to the classical Prim's algorithm for MST construction, where the appearance of a new bar $(0, \lambda)$ corresponds to adding an edge of length $\lambda$ to the MST.

# B   Algorithm for computing the PHD

In Section 4 of the paper, we describe a general scheme for the computation of persistence homology dimension (PHD). Here we give a more detailed explanation of the algorithm.

**Input:**  a set of points $S$ with $|S| = n$.

**Output:**  $\dim_{PH}^0(S)$.

1. Choose $n_i = \frac{(i-1)(n-\hat{n})}{k} + \hat{n}$ for $i \in \overline{1, \ldots k}$; hence, $n_1 = \hat{n}$ and $n_k = n$. Value of $k$ may be varied, but we found that $k = 8$ is a good trade-off between speed of computation and variance of PHD estimation for our data (our sets of points vary between 50 and 510 in size). As for $\hat{n}$, we always used $\hat{n} = 40$.

2. For each $i$ in $1, 2, \ldots k$

   (a) Sample $J$ subsets $S_i^{(1)}, \ldots S_i^{(J)}$ of size $n_i$. For all our experiments we took $J = 7$.

   (b) For each $S_i^{(j)}$ calculate the sum of lengths of intervals in the 0th persistence barcode $E_0^1(S_i^{(j)})$.

   (c) Denote by $E(S_i)$ the median of $E_0^1(S_i^{(j)}), j \in \overline{1, \ldots J}$.

3. Prepare a dataset consisting of $k$ pairs $D = \{(\log n_i, \log E(S_i))\}$ and apply linear regression to approximate this set by a line. Let $\kappa$ be the slope of the fitted line.

4. Repeat steps 2-3 two more times for different random seeds, thus obtaining three slope values $\kappa_1, \kappa_2, \kappa_3$, and take the final $\kappa_F$ as their average.

5. Estimate the dimension $d$ as $\frac{1}{1-\kappa_F}$

# C   Additional experiments

## C.1   Choice of parameters in the formula for PHD

As we have mentioned in Section 3, we estimate the value of persistent homology dimension from the slope of the linear regression between $\log E_\alpha^0(X_{n_i})$ and $\log n_i$. Thus, the exact value of PHD of a text actually depends on the non-negative parameter $\alpha$. The theory requires it to be chosen to be less than the intrinsic dimension of the text, and in all our experiments we fix $\alpha = 1.0$.

Figure 8 shows how the exact value of the PHD for natural and generated texts (of approximately the same length) depends on the choice of $\alpha$. Setting $\alpha = 1.0$ seems to yield reasonable performance, but further investigation of this issue is needed.

For $\alpha \in [0.5; 2.5]$ our results, in general, lie in line with the experiments by Birdal et al. [2021], where performance of $\dim_{PH}^0$ with $\alpha$ varying between 0.5 and 2.5 was studied on different types of data.

## C.2   Effect of paraphrasing on intrinsic dimension

As we show in Section 4, using paraphrasing tools has little effect on the PHD-based detector's ability to capture the differences between generated and natural texts. Here we show how such tampering

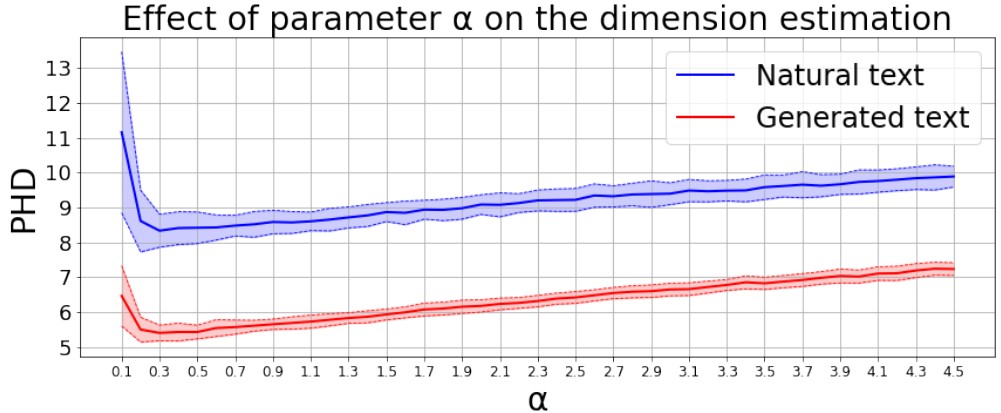

Figure 8: PHD estimates at various $\alpha$ for a natural and a generated texts.

Table 5: Effect of paraphrasing on the intrinsic dimension of the text. Here we can see an uncommon example of data where MLE and PHD behave differently: paraphrasing does not present much trouble for PHD (its quality even increases marginally), but the performance of the MLE-based detector drops significantly, especially for GPT-3.5.

| Method | Original | DIPPER parameters | | | | | |
| | | Lex 20 | Lex 20 Order 60 | Lex 40 | Lex 40 Order 60 | Lex 60 | Lex 60 Order 60 |
|---|---|---|---|---|---|---|---|
| PHD | 7.30 ±1.66 | 7.33 ±1.69 | 7.35 ±1.16 | 7.42 ±1.69 | 7.45 ±1.78 | 7.51 ±1.76 | 7.51 ±1.72 |
| MLE | 9.68 ±1.31 | 9.91 ±1.22 | 9.90 ±1.24 | 9.97 ±1.22 | 9.95 ±1.21 | 10.00 ±1.18 | 10.01 ±1.15 |

with generated sentences affects their intrinsic dimension. Table 5 presents the mean values of PHD and MLE after applying DIPPER with different parameters to the generation of GPT-3.5 Davinci. Where the value of Order (re-ordering rate) is not specified, it means the it was left at the default value 0. Both parameters, Lex (lexical diversity rate) and Order, can vary from 0 to 100; for additional information we refer to the original paper on DIPPER [Krishna et al., 2023].

Increased lexical diversity entails slight growth in both mean PHD and mean MLE that, in theory, should make our detectors less efficient. In the case of a PHD-based detector we do not observe this decrease in performance, probably due to the mean shift being indeed rather small and caused mostly by the right tail of the distribution — the texts that had a high chance of evading detection even before paraphrasing.

Meanwhile, increasing the re-ordering rate from 0 to 60 has almost no noticeable impact.

### C.3  Non-native speaker bias

Here we present full results for our experiments on the bias of ML-based artificial text detectors. Figure 9 presents baseline results for all detectors studied by Liang et al. [2023] that were not included into the main text of the paper.

### C.4  Intrinsic dimension of texts in different languages

Figure 10 presents the PHD of natural and generated texts in different languages on Wikipedia data. Embeddings were obtained from the same multilingual model XLM-RoBERTa-base.

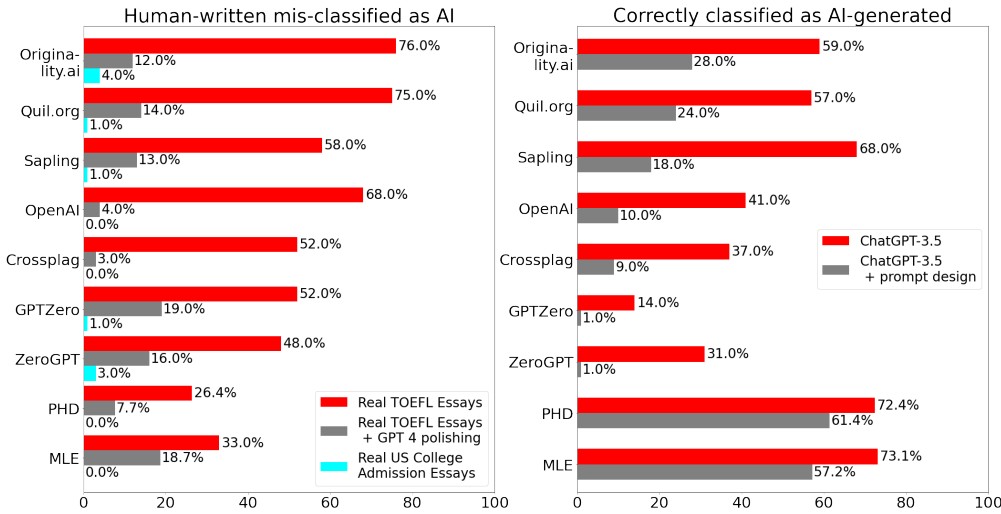

Figure 9: Comparison of GPT detectors in a non-standard environment. Left: bias against non-native English writing samples (lower is better). Right: effect of the prompt design on performance (higher is better).

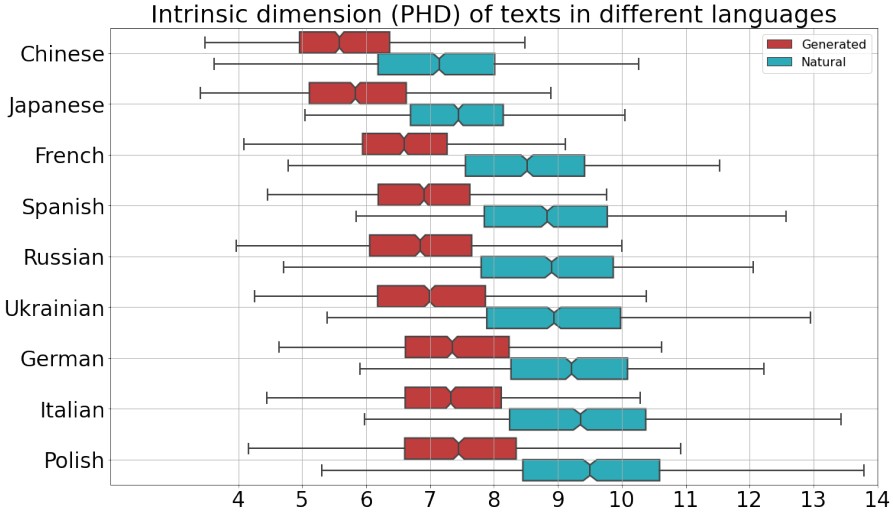

Figure 10: Boxplots of PHD distributions in different languages on Wikipedia data. Embeddings are obtained from XLM-RoBERTa-base (multilingual).

## D   Examples of generated texts

Table 6 provides examples of original text and text generated by ChatGPT.

## E   Examples of misclassified texts

Table 7 provides examples of misclassified human-written texts, while Table 8 shows misclassified texts generated by ChatGPT.

## F   Various Intrinsic Dimension estimators

Table 9 presents average intrinsic dimension estimations obtained by various algorithms for texts of different genres.

Table 6: Examples of original text and text generated by ChatGPT. The common prompt parts are highlighted in bold.

| Original (English) | Generated (English) |
|---|---|
| **USS Mills (DE-383)** 
 **World War II North Atlantic operations** 
 **After shakedown out of Bermuda, Mills trained nucleus crews for frigates and destroyer escorts off Norfolk, Virginia, until 10 January 1944 when she began transatlantic convoy escort duty.** On her second voyage into the Mediterranean, Mills' convoy was attacked before dawn 1 April 1944, 56 miles west of Algiers by German torpedo bombers. SS Jared Ingersoll, a Liberty ship, was hit and set blazing. Mills picked up survivors who had abandoned ship, and sent a boarding party to extinguish her fires. British tug HMS Mindfull and Mills then towed Jared Ingersoll to Algiers. | **USS Mills (DE-383)** 
 **World War II North Atlantic operations** 
 **After shakedown out of Bermuda, Mills trained nucleus crews for frigates and destroyer escorts off Norfolk, Virginia, until 10 January 1944 when she began transatlantic convoy escort duty.** The Mills escorted her first two convoys in January and February from New York City to the United Kingdom, making four round trip crossings by May 1944. During her North Atlantic operations, Mills successfully thwarted several attacks by German U-boats, and carried out rescue operations for nearby ships in distress. She also participated in sweeping the area for mines in preparation for the Normandy invasion. |

| Original (Spanish) | Generated (Spanish) |
|---|---|
| **Currulao** 
 **Instrumentos** 
 **Si la música involucra el uso de una chirimia que es popular en la costa norte, específicamente en el Chocó: Tambor tambora, Cununos ("macho" y "hembra") y un clarinete.** Esta agrupación también puede incluir un eufonio que, en algunos casos, reemplaza o acompaña a un saxofón. Si se trata de un grupo de marimba que es popular en el sur de la región, específicamente al sur del Chocó, Valle del Cauca, Cauca y Nariño: la marimba de chonta, los cununos ("hembra" y "macho"), guasá y el tambor bajo. | **Currulao** 
 **Instrumentos** 
 **Si la música involucra el uso de una chirimia que es popular en la costa norte, específicamente en el Chocó: Tambor tambora, Cununos ("macho" y "hembra") y un clarinete.** El currulao es una danza musical tradicional originaria de la región de la costa Pacífica de Colombia, la cual se deriva de la mezcla de las culturas africana, indígena y española. El sonido del currulao es único y se distingue por el uso de instrumentos que le otorgan su ritmo y cadencia característicos. |

| Original (Polish) | Generated (Polish) |
|---|---|
| **Juan Machuca** 
 **Kariera klubowa** 
 **Całą karierę piłkarską Juan Machuca spędził w klubie Unión Española, w którym występował w latach 1969-1987.** Z Unión Española trzykrotnie zdobył mistrzostwo Chile w 1973, 1975 i 1977. Na arenie międzynarodowej dotarł do finału Copa Libertadores 1975, w którym Unión uległ argentyńskiemu Independiente Avellaneda. | **Juan Machuca** 
 **Kariera klubowa** 
 **Całą karierę piłkarską Juan Machuca spędził w klubie Unión Española, w którym występował w latach 1969-1987.** Był to okres, w którym klub ten odnosił wiele sukcesów, m.in. zdobył mistrzostwo Chile w 1973 i 1975 roku oraz Puchar Chile w 1975 roku. |

| Source, PHD | Text sample |
| --- | --- |
| Human, 5.59 | The route, which is mostly a two-lane undivided road, passes through mostly rural areas of Atlantic and Cape May counties as well as the communities of Tuckahoe, Corbin City, Estell Manor, and Mays Landing. |
| Human, 5.82 | The Meridian Downtown Historic District is a combination of two older districts, the Meridian Urban Center Historic District and the Union Station Historic District. Many architectural styles are present in the districts, most from the late 19th and early 20th centuries, including Queen Anne, Colonial Revival, Italianate, Art Deco, Late Victorian, and bungalow. The districts are: East End Historic District – roughly bounded by 18th St, 11th Ave, 14th St, 14th Ave, 5th St, and 17th Ave. Highlands Historic District – roughly bounded by 15th St, 34th Ave, 19th St, and 36th Ave. Meridian Downtown Historic District – runs from the former Gulf, Mobile and Ohio Railroad north to 6th St between 18th and 26th Ave, excluding Ragsdale Survey Block 71. Meridian Urban Center Historic District – roughly bounded by 21st and 25th Aves, 6th St, and the railroad. Union Station Historic District – roughly bounded by 18th and 19th Aves, 5th St, and the railroad. Merrehope Historic District – roughly bounded by 33rd Ave, 30th Ave, 14th St, and 8th St. Mid-Town Historic District – roughly bounded by 23rd Ave, 15th St, 28th Ave, and 22nd St. Poplar Springs Road Historic District – roughly bounded by 29th St, 23rd Ave, 22nd St, and 29th Ave. West End Historic District – roughly bounded by 7th St, 28th Ave, Shearer's Branch, and 5th St. Meridian has operated under the mayor-council or "strong mayor" form of government since 1985. A mayor is elected every four years by the population at-large. The five members of the city council are elected every four years from each of the city's five wards, considered single-member districts. The mayor, the chief executive officer of the city, is responsible for administering and leading the day-to-day operations of city government. The city council is the legislative arm of the government, setting policy and annually adopting the city's operating budget. City Hall, which has been listed on the National Register of Historic Places, is located at 601 23rd Avenue. The current mayor is Percy Bland. Members of the city council include Dr. George M. Thomas, representative from Ward 1, Tyrone Johnson, representative from Ward 2, Fannie Johnson, representative from Ward 3, Kimberly Houston, representative from Ward 4, and Weston Lindemann, representative from Ward 5. The council clerk is Jo Ann Clark. |

Table 7: The most extreme (outlier) examples of misclassified texts from humans

| Source, PHD | Text sample |
| --- | --- |
| davinci, 31.06 | Moorcraft and McLaughlin's comment reflects the difficulty that the Rhodesian airmen had in distinguishing innocent refugees from guerilla fighters, as the camp was a mix of both. Sibanda's description of the camp further emphasizes the tragedy of the raid, as innocent children were among those killed. |
| davinci, 20.89 | The average annual rainfall at Tikal is approximately 1,500 mm (59 inches). However, due to the unpredictable nature of the climate, Tikal suffered from long periods of drought. These droughts could occur at any time before the ripening of the crops, endangering the food supply of the inhabitants of the city. |

Table 8: The most extreme (outlier) examples of misclassified texts from davinci-003

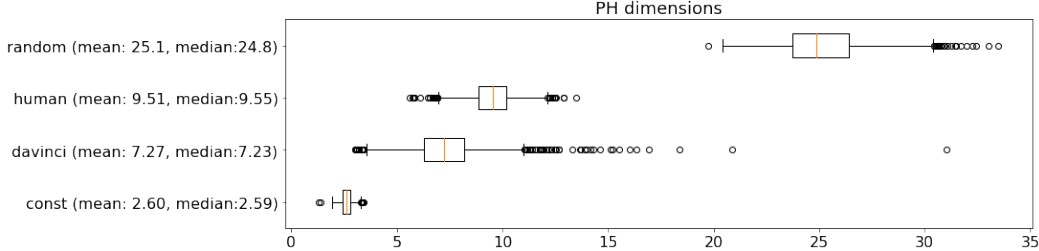

Figure 11: PHD of several types of texts: created by sampling 512 random tokens uniformly from the vocabulary; written by humans (Wikipedia); generated by text-davinci-003 with default temperature (Wikipedia domain); created by repeating the same randomly chosen token 512 times. We didn't perform any additional restarts for outliers correction here for the purpose of highlighting the edge cases. For this reason, one can see more outliers here than on Figure 3. The outlier texts are shown in Table 8

Table 9: Intrinsic dimensions of English texts of different genres estimated by different methods.

| Intrinsic dimension estimator | Wikipedia articles | Fiction stories (Reddit) | Question answering (Stack Exchange) |
|---|---|---|---|
| CorrInt [Grassberger and Procaccia, 1983] | $10.05 \pm 0.57$ | $8.30 \pm 1.64$ | $9.92 \pm 0.71$ |
| FisherS [Albergante et al., 2019] | $6.70 \pm 0.36$ | $6.54 \pm 0.71$ | $7.10 \pm 0.37$ |
| KNN [Carter et al., 2010] | $6.70 \pm 0.36$ | $6.54 \pm 0.71$ | $7.10 \pm 0.37$ |
| lPCA [Cangelosi and Goriely, 2007] | $23.18 \pm 3.60$ | $30.59 \pm 0.71$ | $7.10 \pm 0.37$ |
| MADA [Farahmand et al., 2007] | $366.3 \pm 1410.5$ | $23.2 \pm 83.8$ | $420.7 \pm 960.2$ |
| MOM [Amsaleg et al., 2018] | $11.05 \pm 1.13$ | $5.48 \pm 2.79$ | $12.12 \pm 0.47$ |
| MLE [Levina and Bickel, 2004] | $11.83 \pm 0.77$ | $11.55 \pm 1.20$ | $12.13 \pm 1.00$ |
| **PHD (proposed method)** | $9.49 \pm 1.01$ | $9.21 \pm 1.29$ | $9.59 \pm 1.29$ |
| TLE [Amsaleg et al., 2019] | $10.65 \pm 0.49$ | $10.03 \pm 1.64$ | $11.08 \pm 0.75$ |
| TwoNN [Facco et al., 2017] | $5.85 \pm 1.08$ | $5.70 \pm 1.72$ | $5.95 \pm 1.19$ |

Table 10: Sizes of data splits (# of natural/generated text pairs) used in our experiments on cross-model and cross-domain performance as well as for comparison of our method with universal detectors.

| | Source | Train | Validation | Test |
|---|---|---|---|---|
| Wikipedia | Human/GPT2 | 2001 | 275 | 281 |
| | Human/OPT | 1925 | 277 | 273 |
| | Human/GPT3.5 | 1798 | 259 | 260 |
| Reddit | Human/GPT3.5 | 1677 | 232 | 235 |
| StackExchange | Human/GPT3.5 | 2100 | 300 | 300 |

## G   Data description

Table 10 shows the sizes of data splits that were used to compare our method with universal detectors as well as for evaluating its cross-model and cross-domain performance.

Data for Wikipedia and Reddit was taken from [Krishna et al., 2023]; as for StackExchange, we assembled the dataset ourselves following the same scheme as Krishna et al. [2023]. For all text generation models (GPT2-XL, OPT, GPT-3.5), we randomly selected 2700 pairs of natural/generated texts from the raw data.

First, we divided the data into train/validation/test splits in proportion 7:1:1 to ensure that no human-written texts and no texts generated from the same prompt belong to the train and test split simultaneously. Then we filtered out all texts that were too short for stable estimation of the intrinsic dimensionality (shorter than 50 tokens) and an equal amount of texts from the opposite class.

