# OpenReview forum: "Intrinsic Dimension Estimation for Robust Detection of AI-Generated Texts"
_NeurIPS.cc/2023/Conference — NeurIPS 2023 poster_

### Official Review · Reviewer_98X7 · 2023-07-03

**Soundness:** 3 good
**Presentation:** 3 good
**Contribution:** 3 good
**Rating:** 7
**Confidence:** 3

**Summary:**

This paper proposes a method to detect AI-generated texts based on intrinsic dimensions of sequences generated by humans and LLMs. The authors use a method called persistence homology dimension (PHD) to estimate intrinsic dimensions of both human- and LLM-generated texts. Using a variety of datasets across multiple languages, the authors first notice that intrinsic dimensions between human- and machine-generated texts tend to differ, with the intrinsic dimension of human texts being higher than that of machine-generated ones (the intrinsic dimensions again differ between languages, yet they are always differentiable from human texts).
Using the estimated dimensions, the authors then train a simple single-feature logistic regression classifier to differentiate between human- and machine-generated texts. Experimental results reveal that their method substantially surpasses existing baselines (e.g., DetectGPT, GPTZero), and interestingly the method remains robust against paraphrase attacks conducted using DIPPER.
Finally, the authors show that their method is more robust against samples authored by non-native speakers than existing baselines.

**Strengths:**

* The authors propose a novel method to detect AI-generated content, outperforming existing works.
* The method is effective yet easy to understand, and I believe researchers working on similar topics would be quite interested in the presented results.
* The additional experiments showing that the model is robust against paraphrase attacks further strengthen the method’s relevance.


**Weaknesses:**

* The analysis could be more extensive. For example, given that the classifier is based on a single feature, it would be interesting to see how performance varies as a function of dataset size, and how vulnerable its generalization capabilities are to ‘noisy’ datasets (i.e., those where dimensions for individual samples have high variance).  While this has partially been explored (Figure 2), showing classification results in the Experiments section would have been informative.
* It would be nice if Table 1 could be extended for additional approaches to ID estimation (line 188 mentions that 12 approaches have been explored). Showing results across those approaches (e.g., in a boxplot) would give the reader a better notion of how well the differences in dimensions between human- and AI-generated texts generalize across methods.


**Questions:**

None

**Limitations:**

The authors clearly outline the limitations of their work in a dedicated section. However, a few additional words on the broader impact (e.g., the applicability of the method in practice) would be helpful.

---

> ### Author Rebuttal · Authors · 2023-08-09
>
> Thank you for your review!
>
> We have considered adding more analysis of ID estimations for the texts to appendices. In fact, we have explored some types of "linguistic noise", i.e., deviations from standard language. In particular, we use data from Reddit that contains quite informal texts (with a notable amount of colloquialisms) as one of our primary datasets; the standard deviation of ID estimates for them is ~28% higher than for Wikipedia texts, but the mean is still the same. We also analyze the performance of our method on texts produced by non-native speakers, and it also shows a higher variance of ID estimates. We leave more detailed investigations of the impact of other kinds of noise in data for future work.
>
> Due to space limitations, we could not include the data for other ID estimation approaches in the primary text, so we limited Table 1 to two methods only. Thank you for this comment; we will add an extended version of Table 1 (with other ID estimators) into the appendix.

---

> > ### Comment · Reviewer_98X7 · 2023-08-15
> > **Acknowledgement of rebuttal**
> >
> > Thanks to the authors for addressing my concerns. I would encourage you to add the details provided in your response to the paper (as you have suggested). As already indicated with my initial scores, I believe this is solid work and therefore recommend acceptance.

---

### Official Review · Reviewer_NYsD · 2023-07-05

**Soundness:** 4 excellent
**Presentation:** 3 good
**Contribution:** 4 excellent
**Rating:** 7
**Confidence:** 4

**Summary:**

The paper describes work on detecting AI-generated text using intrinsic dimensionality (ID) estimation methods through Persistent Homology Dimension (PHD) (and MLE).  The authors motivate this approach by highlighting that written texts between machine and humans observe differences in topological representation. The authors explored this method using an extensive set of experiments covering various SOTA models for detection, such as DetectGPT, GPTZero, etc., across multiple languages in a crosslingual setup and across multiple source domains of data such as Reddit, StackExchange, and Wikipedia. Results show that using ID as a feature generally performs consistently and somewhat robustly across other methods, especially in general detection.

Overall, I believe the paper is a good addition to NeurIPS, especially as it gives a unique perspective on detecting machine-generated texts that’s theoretically motivated.


**Strengths:**

There are a number of things that I liked about the paper that contributed to my overall recommendation. First, the paper gives a unique point-of-view to the standard AI-generated text detection task which deviates from the usual approach of looking at surface-based style and linguistic characteristics. The paper shows strong motivation and support as to why intrinsic methods such as PHD shows substantial differences between human-written and machine-generated texts. Second, I find merit in the breadth and depth of the experiments conducted which extends to crosslingual and crossdomain settings. It is also good that the authors explored a number of generative models and targeted some of the concerns with detecting machine-generated text such as errors with texts from non-native speakers. Lastly, the paper is readable enough for a larger variety of readers to appreciate its contributions. With these points, I recommend the paper for acceptance.


**Weaknesses:**

There are no major issues in the paper that I’m particularly concerned about. There are some points, however, that I would recommend to be given more emphasis and clarification to improve the quality of the discussion as well as to increase confidence with the results.

1. I would have appreciated it better if the authors provided their clear inferences as to why the PHD-based detector, even with a paraphraser, is substantially better than the other classifiers. If the result indeed shows that the method is effective, what does it imply? Does it suggest that using a non-linguistic representation such as the intrinsic dimension of data compared to other linguistic factors considered by DetectGPT, RankGen, etc. is more practical?  Quantitatively, how much offset in performance do paraphrasers like DIPPER affect the proposed classifier as well as the other detectors? It would also be interesting and a good addition in the discussion if the authors can show that even using paraphrasers, the proposed detector is still significantly effective through a statistical test.

2. There are a few vague statements in particularly focused on parts of the discussion of results. In the cross-domain and cross-model performance, statements such as “the PHD classifier is not influenced by domain transfer” and “PHD classifier to be more robust to entirely new AI models” need further clarification and even rephrasing. Achieving relatively favorable performance on three datasets hardly makes any model robust and immune to domain shift unless proven in-depthly with more data and/or backed by statistical tests.

3. It seems odd that ChatGPT was never mentioned anywhere in the statement of models used for experiments and then appears on one of the results section. Was this portion rushed? Justification is needed for this as readers would expect all experiments on evaluating generations to come from what was previously declared and should be uniform for all experiments. Also, there’s no “GPT3.5-davinci-003” in OpenAI’s model endpoints (https://platform.openai.com/docs/models/model-endpoint-compatibility). This may have been a typo so kindly clarify.

4. Same recommendation also applies where the authors should provided more discussion instead of describing the values on the table as seen in the experiment on non-native written text vs. AI. It was observed that existing models such as GPTZero and OpenAI tend to have higher rates of misclassifying non-native text as AI-generated but not so much for PHD and especially using MLE-based classifiers. Thus, what are the author’s insights as to why MLE performs better than PHD for this type of experiment? The authors previously mentioned that MLE performs well on “small samples, noisy settings, and small variance” so how does this ties up with the result? What gives MLE the edge over PHD in this scenario?

5. There are mentions of using specific splits of existing datasets in the paper. These should all be added as a separate table in the Appendix. It’s quite challenging as the reader to approximate how large the data used specifically for the study. This information is also used when analyzing the model performances.


**Questions:**

Aside from the supporting questions highlighted in the Weaknesses section, here are some points that further require clarification:

1. Is it scientifically correct to refer to the intrinsic dimensionality methods as non-linguistic approach for detecting AI-generated texts? To me, the method seems to merit its own category compared to approaches using stylistic properties of texts.
2. What decoding hyperparameter values were used? Are these uniform across all model generation setups?
3. Is it possible that the training setup of RoBERTa contributed to it being more robust across various domains? RoBERTa’s performance was highlighted but the authors did not provide any of their own inferences.


**Limitations:**

The added limitations section seems acceptable but I would recommend to further clarify the breadth of dataset used to merit claiming robustness and applicability to domain shift. It might be worth comparing to previous literature on how much data they used before they can refer to one model as robust.

---

> ### Author Rebuttal · Authors · 2023-08-09
>
> Weaknesses.
>
> 1. We suppose that modem language models can mimic human-written texts very well in terms of common linguistic properties such as grammar, semantic, style etc., but there are subtle differences that can be captured via numerical analysis of the topology of text embeddings or the curvature of the probability function, somewhat similar to the approach of DetectGPT. We consider it as an important and interesting direction for future work.
>
> 2. Naturally, we cannot state that our method will work in any domain shift situation. Our point was to demonstrate that ID of the text embeddings is a reliable characteristic of text that helps to discover the above-mentioned subtle differences in text data and disentangle them from text style and semantics.
>
> 3. Thank you for your remark! By “GPT3.5-davinci-003” we meant the model “text-davinci-003” from /v1/completions. By “ChatGPT” we meant the model  “gpt-3.5-turbo” from /v1/chat/completions. We often refer to both models as “GPT-3.5” or “GPT-3.5 family” in the introduction and other parts of the paper because both are listed as such in the list of  “GPT-3.5 models” at https://platform.openai.com/docs/models/gpt-3-5 . If accepted, we will review our naming and make it more clear and precise in the text (i.e., replace “GPT3.5-davinci-003” by “text-davinci-003”; specify more clearly that we refer to both davinci and ChatGPT as “GPT-3.5” in the introduction etc.).
>
> 4. Table 4 shows that PHD and MLE have roughly the same ROC-AUC for ChatGPT texts in English. The higher accuracy on 1% FPR means that MLE provides a shorter left “tail” on real data, i.e. there are fewer real texts with very low MLE. We suppose that this is exactly because it works better with small samples. As we have mentioned in the answer to Reviewer 1, many of the PHD extreme cases are too short.
>
> 5. Thank you for pointing this out. We planned to upload exact splits into the Github repository with the code for this paper, but we can also add tables with sizes of used splits to the appendices.
>
> Questions
>
> Q1. Our approach can be called “non-linguistic” in the sense that we are not working with text or language *directly*; instead, we analyse its mathematical representation (which, of course, inherits various features of text).
>
> Q2. Yes, they are uniform for every model across all generation setups. For our dataset, we used ChatGPT with default parameters (temperature 0.7). We used texts generated by GPT-2, GPT-3.5, and OPTb that were published by Krishna et al. (authors of "Paraphrasing evades detection..."), and detailed information on the choice of hyperparameters can be found there. We followed the procedure from the paper, and in experiments with the intrinsic dimension reported results on data without watermarking (strength_0.0).
>
> Q3. Indeed, RoBERTa (which is exactly “Robustly optimized BERT”) is developed to be robust. The important factor which is shown to improve the performance on downstream tasks is increasing the training dataset by an order of magnitude and, importantly, adding more variability to the data. While BERT was trained on Books and Wiki only, RoBERTa was fed also with a large web crawled corpus. So, both Reddit and StackOverflow are the types of data seen during pretraining. We believe that this is, indeed, the root cause of the robustness of ID estimation for RoBERTa embeddings. We suppose that ID estimation is more stable for data familiar to the embedding model. We plan to test this hypothesis in our further study.

---

> > ### Comment · Reviewer_NYsD · 2023-08-12
> > **Response acknowledged. Score remains the same (Accept)**
> >
> > This is to acknowledge that I have fully read both my fellow reviewers' feedback as well as the author/s' response to my own assessment. For the rebuttal period, the authors have provided some additional information for my concerns listed in my review (see weaknesses section). With this, I would like to summarize some points that the authors are strongly recommended add to the final paper in case of acceptance. This will ensure that any claims made on the paper are properly supported with evidence as well as discussed thoroughly.
> >
> > 1. In response to #1 weakness, the authors say that *"there are subtle differences that can be captured via numerical analysis of the topology of text embeddings"*. This is still vague, in my opinion. As suggested, it would be better to show a statistical test of difference on a number of runs with the ID method (say with different splits) for Table 2 or in Figure 6, with or without paraphrasers.
> > 2.  In response to #2 weakness, the authors say that *"we cannot state that our method will work in any domain shift situation."*, however when you read the paper, there are mentions of *“the PHD classifier is not influenced by domain transfer”* and *“we can expect the PHD classifier to be more robust to entirely new AI models”*. I suggest rephrasing these sentences instead, as it gives an inflated expectation for the classifier model.
> > 3. Response to #4 weakness and Question #3 are good additions in their appropriate sections in the paper given provided more details. I was specifically looking for these in the paper.
> >
> > Nonetheless, I would like to thank the authors for their efforts and for being very detailed in their responses to my questions and concerns.  My score will remain the same (7 - Accept) and would be happy to vouch for the paper to be accepted to the conference.

---

### Official Review · Reviewer_oVCJ · 2023-07-06

**Soundness:** 3 good
**Presentation:** 2 fair
**Contribution:** 3 good
**Rating:** 7
**Confidence:** 3

**Summary:**

This paper proposes a new method for artificial text detection(ATD) with intrinsic dimension (ID) estimation. First, contextualized representation of tokens in the text is extracted by a RoBERTa model. Next, the author estimates the ID of this set of contextualized representations : (1) N tokens and their corresponding vectors are sampled from the set, forming a vertex set; (2) the persistent homology dimension (PHD) is estimated by measuring the lengths of the edges in a minimal spanning tree with the vertex set; (3) varying the value of N to obtain a set of N-PHD pairs, and measuring the ID based on the slope of their linear correlation. Finally, the ID is utilized as a feature in a logistic regression model for binary classification.

The author has conducted extensive experiments on widely-used benchmarking datasets. The results demonstrate that the proposed method exhibits greater robustness when compared to existing ATD methods, particularly in terms of its resilience towards adopted AI models, paraphrase attacks, and non-native speakers. Furthermore, in comparison to a conventional RoBERTa-based classifier, this method showcases significantly improved out-of-domain performance in both domain and AI model transfer scenarios. Additionally, the author has curated a new dataset for multilingual ATD, although specific details regarding this dataset are not provided.


**Strengths:**

1. The application of persistent homology and intrinsic dimension estimation for ATD is both intriguing and well-founded in theory. While the methodology is similar to that of [1], which is already cited in the paper, this method relies on fewer features and conducts more comprehensive experiments to assess the robustness of competing methods. Hence, the proposed method and its findings exhibit a significant level of novelty.

2. The proposed method has achieved promising performance, particularly in terms of robustness. This is mainly because it does not need to finetune parameters of a large Transformer model, nor does it assume reliance on an AI model that is likely to generate the text.

3. Extensive experiments and analysis are done to demonstrate the effectiveness of the method.

[1]  Kushnareva, Laida, et al. "Artificial Text Detection via Examining the Topology of Attention Maps." EMNLP. 2021.


**Weaknesses:**

1. Some claims are not well supported. The author claims that they estimate the intrinsic dimensions (ID) of text data. In fact, what they really estimate is the ID of the contextualized representations. By contrast, [1] directly estimates the ID of natural images.

2. Although extensive experiments have been conducted, one important analysis is missing: investigating the impact of the decoding algorithms. One important assumption of this work is that the metric space of human-written text has more isolated sub-graphs, which can be incurred by rapid shift of topic, usage of rare words, and so on. AI models can also achieve these characteristics by adjusting the hyperparameters of decoding algorithms, such as reducing the softmax temperature or increasing the values of P and K in nucleus and top-K sampling, respectively. As seen in Table 2, PHD has better performance for detecting GPT-3.5 than GPT-2, mostly because GPT-2 tends to generate text that has more grammar mistakes and less akin to human language. Consequently, the reliability of the method in the face of different sampling techniques becomes questionable.

3. The paper is not very easy to follow: some important details and definitions are missing. To name a few:
   * Some background information about topological features, especially connected components, are necessary to understand the estimation of persistent homology dimension.

   * Lack of definition of $C$ in Line 177, as well as $\tilde{C}$ in Line 183.
   * In Section 3, the author solely explains the estimation process for $E^{i=0}$ without explicitly indicating that they set $i=0$ until Section 4. It would be clearer if this information was explicitly stated earlier.

   * The author lists the new dataset as one of the main contributions of their work. However, no specific details are presented. It would be helpful to know the dataset's statistics, the curation process employed, as well as any annotation, cleaning, or pre-processing steps that were carried out.



[1] Pope, Phil, et al. "The Intrinsic Dimension of Images and Its Impact on Learning." ICLR. 2020.


**Questions:**


## Typos

1. Line 45: some downstream task(s)
2. Line 177: where equivalence mean(s)

## Suggestions

1. It would be helpful to include some text examples with high and low ID.

## Questions

1. Why RoBERTa-large has higher variance than RoBERTa-base(Figure 5)
2. What does the * mean in Table 2?
3. Why do you shuffle the order of datasets in Table 3? It is weird that PHD has higher performance on OOD than ID.
4. Why https://arxiv.org/abs/2104.08894 report much higher dimension for images?
5. Line 177: “equivalence mean that…” where is the equivalence?
6. Caption in Figure 6 is confusing: the title of the right-hand-side graph says it is the success rate of classification (higher the better) but the caption says it is the decrease of performance (lower the better).
7. What is the ID of the text generated by randomly sampling tokens from the vocabulary?



**Limitations:**

The limitations are fairly elaborated.

---

> ### Author Rebuttal · Authors · 2023-08-09
>
> Thank you for your review!
>
> First, we address the “Weaknesses” section.
>
> 1. We use the notion “ID of text” for simplicity; in contrast to image processing, text embeddings are the main numerical representation of texts widely used in NLP nowadays, so we believe there is very little chance of confusion caused by this inaccurate terminology. Text data, regardless of the type of embeddings, has common properties, e.g., it is discrete, and there is a relatively small number of points available for each text. Our methods of ID estimation are able to deal with such types of data.
>
> Experiments of this work are restricted to text embeddings obtained by Transformer MLMs. But our focus is not on the properties of these embeddings; we show that embedding sets obtained by the same model have different estimated IDs for different types of data, and we study the properties of data via their embeddings’ ID. To be precise, we estimate the ID of a set of embeddings obtained by RoBERTa-family models, for both natural and generated texts, and show that they differ.
>
> We thank you for pointing out this confusion and will clarify it in the text if accepted.
>
> 2. Unfortunately, space and time constraints precluded us from performing a thorough analysis of the impact of the decoding algorithm parameters and including the results of this analysis in the paper. We will include them into the appendices after concluding additional experiments. We can note here that ID estimation of texts obtained from GPT-3.5 exponentially depends on temperature: texts obtained with temperature values up to 1.2 usually have lower ID than human-written texts, but if temperature is at least 1.7 then generated texts have ID notably larger than human-written ones. As for GPT-2, we found that top-K sampling (with K=40) on average leads to ~15% lower values of ID.
>
> 3. We thank you for noting unclear details in the paper. We will fix them and add the necessary explanations in the final version if accepted.
>
> 4. Thank you for this suggestion! We will definitely add this information to the paper if accepted.
>
> Questions
>
> Suggestion 1.
>
> Please find some examples in Table 1 in the attached file. It seems that most examples with lower ID contain a lot of addresses, geographical names, or proper nouns. It may be connected with the usage of rare tokens or with more numbers in the text than usual, but more experiments are needed to be certain.
>
> Some examples, however, are just very short texts, for which our method of PHD calculation is less reliable. This means that there is high variability of the ID values for the *single* sample, calculated with different random seeds. On our estimation, this kind of data occupy the  lowest and highest 0.5% of the distribution.
>
> Q1. We hypothesize that it may be caused by larger embedding size in RoBERTa-large (1024 compared to 768 for RoBERTa-base). It is known from literature that increasing the latent space dimension may often negatively affect the stability of individual ID estimates for fractal-based methods [1] (and PH-dim is one of such). See: Camastra and Staiano, Intrinsic Dimension Estimation: Advances and Open Problems, Information Sciences, vol. 328, pp. 26-41, 2016.
>
> Q2. * means that this number is not directly comparable with other numbers in the table because the detector uses weights of exactly the same model that was used for generation, so it does not belong to the class of universal detectors. This issue for the pair DetectGPT-GPT2 is mentioned in the subsection “Comparison with universal detectors” (l. 262-263 and 268-269). We will add this explanation to the table caption if accepted.
>
> Q3. This is a typo in table heading; thank you for pointing it out. The correct order of labels is given in the left half of the table: Wikipedia, Reddit, StackExchange. Meanwhile, numbers are given in the correct order.
>
> Q4. Thank you for the useful reference!
> The setup of this paper differs from ours. We estimate the dimensionality of each text separately; they estimate the dimensionality of the entire dataset considering every image as a separate point. Naturally, such estimation yields larger numbers with larger variation.
> Table 1 and 2 and Figure 1 in that paper show that ID estimation depends both on the dataset and estimator properties, and varies from 7 to 45, while our estimation mainly lies in the 7-12 range. As mentioned above, our estimation also depends on the properties of the embedding model.
>
> Q5. It is related to the equation from the same line: $E^0_\alpha(X) \sim C n^{\frac{d-\alpha}{d}}$, here $\sim$ is read as “equivalence”. We will state it more clearly in the final version if accepted.
>
> Q6. We opted to preserve the style of figures from the paper where those experiments were originally introduced. The plots themselves are correct (lower is better on the left-hand size, while higher is better on the right-hand side); we will clarify it in the caption of the figure.
>
> Q7. In Figure 1 of the attached file, we show boxplots of IDs of texts made up of repetitions of the same tokens (const), generated by davinci, written by humans, and generated by sampling random tokens from the RoBERTa vocabulary. It supports our hypothesis that embeddings of more homogenous texts tend to have lower ID, while embeddings of more heterogeneous texts tend to have higher ID.

---

> > ### Comment · Reviewer_oVCJ · 2023-08-18
> >
> > I have read the response, and I am still leaning towards accepting the paper.

---

### Official Review · Reviewer_SSuo · 2023-07-07

**Soundness:** 3 good
**Presentation:** 3 good
**Contribution:** 3 good
**Rating:** 5
**Confidence:** 3

**Summary:**

The paper proposes  using the intrinsic dimensionality of the manifold underlying the set of embeddings of a given text to detect AI-generated texts. Because the average intrinsic dimensionality of AI-generated texts is lower than that of  natural language. It is found that the intrinsic dimensionality of different languages varies a lot.

**Strengths:**

- The paper itself is novel
- The finding that  the intrinsic dimensionality of different languages varies a lot is interesting.
- The method is robust in cross-domain and cross-model scenarios.

**Weaknesses:**

- The paper does not discuss the reason for the difference of intrinsic dimensionality in different languages.
- The reason why the method works isn't clearly understood.

**Questions:**

- Intrinsic dimensionality is calculated in BERT/RoBERTa emebedding, I wonder if the *syntax of individual language* leads to the *difference of intrinsic dimensionality in different languages*.
- Is there any analysis on the bad cases (incorrectly-classified cases)?  which might help better understand the method.


**Limitations:**

See the Weaknesses

---

> ### Author Rebuttal · Authors · 2023-08-09
>
> Thank you for your review.
>
> In our work, we study Intrinsic Dimensionality estimation for text embeddings and show experimentally that this mathematical value can reflect some useful information about the given text, namely, helping to separate artificial and human written texts. We admit that a huge amount of important questions are left out of the scope of this paper. To the best of our knowledge, we are the first to show that ID estimation of text embeddings applies to this kind of downstream task, and just couldn’t cover all the questions related to this method in a single paper.
>
> One of the most important questions is the one raised by Reviewer SSuo: why does this method work? Indeed, this has not been clearly understood yet. There is mathematical intuition behind the notion of intrinsic dimensionality; using it, we can conclude that GPT-generated texts have a smaller number of degrees of freedom, i.e. less creative in some sense. We believe that there are some subtle semantic, syntactic, stylistic, or statistical differences, which are reflected by ID value and could support this intuition. Unfortunately, so far, we could not discover these properties; but our work proves experimentally that some kind of differences exist. We hope that future research will shed some light on this question.
>
> Answer to Q1. The difference between the ID of different languages could be caused by many reasons: the properties of the language, e.g. analytical vs. synthetic; the script (e.g. syllabic or letter); the quality of the language representations of the embedding model; the quality of the data (e.g. Wikipedia articles can have different quality for different languages), etc. Interesting to note that in Fig. 4 Asian languages have smaller ID than European; among European, related languages are often grouped (Russian and Ukrainian, Spanish and Italian). We can hypothesize that geographical and linguistic connections between languages lead to similar ID; this hypothesis should be a subject of accurate investigation, and we left it for future research.
>
> Answer to Q2. As for bad case analysis, we noticed an interesting tendency among the examples with the lowest ID of human-written texts (those will be surely misclassified). It seems that most of these examples contain a lot of addresses, geographical names, or proper nouns. It may be connected with the usage of rare tokens or with more numbers in the text than usual, but more experiments are needed to claim it for sure. Some examples, however, are just very short texts. Examples are provided in Table 1 in the attached file.
> We didn’t notice any significant anomaly in the top ten examples with the highest ID of Davinci-generated texts, except that all these texts were very short. We hypothesize that PH dimension estimation doesn’t work properly on short text. More thoughtful analysis of bad cases is a matter for future work.
>
> On Fig. 1 in the attached file we provide extreme cases analysis. The most simple samples have the lowest ID, and the highest ID corresponds to the completely random ones. It corresponds to the general understanding of ID as the amount of degrees of freedom in the data.

---

### Author Rebuttal · Authors · 2023-08-09

We are thankful to all the reviewers for their inspiring reviews! In the attached file, we provide examples and figures illustrating  extreme cases of ID values. We discuss these results in the direct answers to the reviewers (SSuo and oVCJ).

---

### Decision · Program_Chairs · 2023-09-21

**Decision:**

Accept (poster)

**Comment:**

All reviewers unequivocally agreed that this submission to be accepted. One room for improvement the AC is noticing is that most references do not go past 2019. I'm pretty sure the term "intrinsic dimension" in the deep learning era was coined by "Measuring the Intrinsic Dimension of Objective Landscapes" , ICLR 2018.